# Comparison of Binding Properties of a Laccase-Treated Pea Protein–Sugar Beet Pectin Mixture with Methylcellulose in a Bacon-Type Meat Analogue

**DOI:** 10.3390/foods12010085

**Published:** 2022-12-23

**Authors:** Pascal Moll, Hanna Salminen, Lucie Stadtmueller, Christophe Schmitt, Jochen Weiss

**Affiliations:** 1Department of Food Material Science, Institute of Food Science and Biotechnology, University of Hohenheim, Garbenstrasse 21/25, 70599 Stuttgart, Germany; 2Department of Chemistry, Nestlé Institute of Material Sciences, Nestlé Research, Vers-chez-les-Blancs, CH-1000 Lausanne, Switzerland

**Keywords:** biopolymer blend, pea protein, sugar beet pectin, laccase, methylcellulose, binder, binding agent, textured vegetable protein, fat mimetic

## Abstract

A bacon-type meat analogue consists of different structural layers, such as textured protein and a fat mimetic. To obtain a coherent and appealing product, a suitable binder must glue those elements together. A mixture based on pea protein and sugar beet pectin (r = 2:1, 25% *w*/*w* solids, pH 6) with and without laccase addition and a methylcellulose hydrogel (6% *w*/*w*) serving as benchmark were applied as binder between textured protein and a fat mimetic. A tensile strength test, during which the layers were torn apart, was performed to measure the binding ability. The pea protein–sugar beet pectin mixture without laccase was viscoelastic and had medium and low binding strength at 25 °C (F ≤ 3.5 N) and 70 °C (F ≈ 1.0 N), respectively. The addition of laccase solidified the mixture and increased binding strength at 25 °C (F ≥ 4.0 N) and 70 °C (F ≈ 2.0 N), due to covalent bonds within the binder and between the binder and the textured protein or the fat mimetic layers. Generally, the binding strength was higher when two textured protein layers were glued together. The binding properties of methylcellulose hydrogel was low (F ≤ 2.0 N), except when two fat mimetic layers were bound due to hydrophobic interactions becoming dominant. The investigated mixed pectin–pea protein system is able serve as a clean-label binder in bacon-type meat analogues, and the application in other products seems promising.

## 1. Introduction

Plant-based meat analogues have experienced a boost in sales, with new products being developed at a fast pace in recent years. This food product category has grown from a niche product into a well-established category. This is due to consumer perceptions that plant-based meat analogues are better for the environment and may help to promote health and well-being [1,2,3,4]. Most meat analogues aim to mimic the properties of traditional meat products as closely as possible; however, the processes and formulations differ depending on the product category [5,6,7]. For instance, whole-cut meat analogues consist of fibrous chunks that are mainly produced by extruding a protein-rich raw material. Lipids, colorants, and flavors are used as well, but they are not necessarily needed for the coherence of the final product [6]. Meat analogues that resemble so-called ground and bound products, such as burgers, patties, and nuggets, also consist of textured protein pieces; however, they are processed into much smaller particles when compared to whole-cut products and then mixed with other components, such as fat particles or oil droplets. However, textured vegetable protein and fat particles do not necessarily stick together on their own [8]; therefore, an additional component is necessary to hold everything together and create a coherent and appealing product, i.e., binders [6,9]. Furthermore, flavor and color components may be added to imitate sensorial appearance of the original.

Binders function through a variety of mechanisms. In burger patties, binders build a continuous layer around the different structural elements and, upon heating, solidify, thereby creating a matrix with a texture and appearance much like the original meat product [10,11,12]. Methylcellulose solutions are able to reversibly build a gel upon heating [13], making it the most frequently used binder in burger patty analogues to date [5]. For instance, the hardness of plant-based patties made from commercial textured vegetable protein increased when the concentration of methylcellulose acting as binding agent was increased from 1.5 to 4%. However, the hardness still was significantly lower compared to the beef control [10]. Methylcellulose, however, suffers from a low consumer acceptance due to its synthetic nature, and consumers increasingly try to avoid it [6,14]. Furthermore, the binding mechanism in another product category, namely in bacon-type meat analogues, is instead dominated through adhesion and cohesion, i.e. stickiness, as large pieces of textured vegetable proteins and fat mimetic must be glued together [15,16]. Therefore, gelling is of limited importance, and other factors that govern stickiness are of greater importance. Potentially, stickiness and hence the binding strength increases with the ability of the binder to (i) adsorb to the adherend (=textured protein or fat mimetic layer) and exhibit interactions; (ii) to mechanically interlock with the adherend; and/or (iii) to form covalent bonds with the adherend. These are influenced by factors such as the deformability and wetting of the binder, as well as the chemical makeup and reactivity of both the binder and the adherend, and the surface roughness of the adherend [17,18,19,20,21]. Taken together, the binding mechanism in the popular product category of bacon-type meat analogues functions differently when compared to other meat analogue products. Therefore, a bacon-type meat analogue is an interesting model matrix to study the suitability of using a sticky food glue, instead of methylcellulose that binds through gelling.

We recently reported that a concentrated mixture of pea protein and apple pectin exhibits properties that make it suitable to serve as a sticky food glue, when the right balance between adhesion and cohesion is struck [22]. Furthermore, the mixture can be solidified with laccase when the pectin is derived from sugar beet with crosslinks between ferulic acid groups, increasing network strength and cohesion [8]. In addition, laccase treatment can potentially form crosslinks between the binder and the adherend, thereby benefitting adhesion. Such a process-driven transformation of molecular interactions, including a change in functionality, bears great potential for the food industry [23]. Based on those results, we aimed to find a novel, clean-label binding agent that can replace methylcellulose in commercial applications. We hypothesized that a concentrated mixture of pea protein and sugar beet pectin, treated with laccase, may provide sufficient binding between different structural elements (textured vegetable protein and fat mimetic) in a bacon-type analogue. Methylcellulose served as a benchmark binder in our study. 

## 2. Materials and Methods

### 2.1. Materials

High-moisture textured vegetable protein with 30.7% dry matter was provided by DIL (Deutsches Institut für Lebensmitteltechnik e.V., Quakenbrück, Germany) and was made from a soy protein concentrate (ALPHA^®^ 8) from Danisco (Wilmington, DE, USA). An emulsified and crosslinked fat crystal network was prepared according to Dreher, et al. [24] with modifications, and served as a fat mimetic. In brief, the soy protein isolate was thoroughly mixed with a lab rotor-stator homogenizer, adjusted to pH 7, and rapeseed oil was slowly added while homogenization continued. Then, transglutaminase was added; the mass was filled into casings and incubated at 40 °C for 60 min to enable protein crosslinking. Afterwards, the mass was heated to a core temperature of 85 °C for microbial safety and stored at 2 °C until further usage. The ratio of 12% soy protein isolate suspension (Supro Ex 37 HG IP, Solae Europe S.A., Geneva, Switzerland) to rapeseed oil (MEGA eG, Stuttgart, Germany) was set at 40:60 [24]. Pea protein Pisane^®^ C9 (Cosucra, Warcoing, Belgium) had a protein content of 67% (N × 5.36), 10% carbohydrates, 9% moisture, 6% ash, 8% fat [25]. The sugar beet pectin (Herbstreith & Fox GmbH & Co. KG, Neuenbürg, Germany) had a molecular weight of 45.0 kDa and a 54% degree of esterification, according to the manufacturer. Methylcellulose (WELLENCE™ Vege Form 183) was obtained from Danisco (Wilmington, DE, USA). Sodium hydroxide with ≥99% purity, sodium chloride, calcium chloride, and brilliant black BN (C.I. 28440) was purchased from Carl Roth GmbH & Co. KG (Karlsruhe, Germany).

### 2.2. Preparation of Binders

A previously reported method [22] was used to prepare a viscoelastic food glue that was based on pea protein and sugar beet pectin. The pea protein to sugar beet pectin weight ratio was 2:1; the total biopolymer concentration (=pea protein & sugar beet pectin) 25% (*w*/*w*), and 100 mmol of NaCl was added. The pH was adjusted to 6.0 with NaOH, as this was shown to promote stickiness and lies within the pH optimum for laccase [26]. A third of the required water at room temperature was used for dissolving 100 nkat laccase per gram solid substance (25% *w*/*w*) [27], which was then added to the mixture and incorporated into it by stirring vigorously by hand, until a homogeneous mass was obtained. This happened right before use of the binder for assembly of the bacon-type meat analogue (see Section 2.4). A sample without laccase addition served as a control. 

Methylcellulose hydrogels were prepared by weighing 2–12% (*w*/*w*) methylcellulose powder in water and stirring it in a food processor (UMC 5, Stephan, Diessen, Germany) at medium speed for 10 min. A vacuum was applied during stirring with a diaphragm vacuum pump (MZ 2C, Vacuubrand, Wertheim, Germany) to avoid the incorporation of air bubbles. The native pH (7.4 ± 0.1) of the gels was not adjusted, as we have seen in preliminary experiments that the pH had no influence on the stickiness of a methylcellulose hydrogel.

For all binders, sodium azide (0.02% *w*/*w*) was added to prevent microbial growth during storage, and brilliant black (0.01% *w*/*w*) was added to dye the binder for better visualization.

### 2.3. Probe Tack Test

Probe tack tests were performed with a texture analyzer (TA-X12, Stable Micro Systems, Godalming, UK) to evaluate the stickiness of the methylcellulose hydrogels that were placed in small glass vessels (diameter = 3 cm, height = 1 cm). A cylinder probe (d = 25 mm) was driven downwards at 0.5 mm/s until it reached the sample. The sample was slightly compressed at 0.392 N for 0.1 s, which triggered a quick back-movement process of the cylinder probe (at 10 mm/s) at a distance of 10 cm. Measurement took place at 25 °C, and the failure mode after the test was determined.

### 2.4. Tensile Strength Test

A tensile strength test was adopted with modifications [15]. The binders (20 mg cm^−2^) were applied on a circular-shaped piece of textured vegetable protein (TVP) with a diameter of 4.5 cm and a height of 0.4 cm. Then, another piece of TVP or fat mimetic of the same dimensions was put on top. Two fat mimetic pieces could only be glued together with methylcellulose, since the application of pea protein–sugar beet pectin mixture on the fat mimetic directly led to rupture. The lower piece of the sandwich-like structure was glued with cyano-acrylate glue (Superglue 80853, BGS technic KB, Wermelskirchen, Germany) into a holding device. A metal stamp with a weight of ca. 100 g was glued on the upper piece of the sample. All samples were wrapped in aluminium foil to avoid dehydration during the next steps. The samples with pea protein-pectin mixture as binder were incubated in a heating oven (Euromat Kombi-Dämpfer, Wiesheu GmbH, Großbottwar, Germany) at 45 °C for 4 h, during which a core temperature of 40 °C was reached to facilitate laccase action. Preliminary experiments showed that those parameters allowed for sufficient solidification. For the samples with methylcellulose hydrogel as binder, this incubation step was omitted. Then, a step to inactivate the enzyme in all samples was carried out at 95 °C for 17 min, with the core temperature remaining at 90 °C for at least 5 min. The samples were mounted into the texture analyzer (Instron Model 3365, Instron Engineering Corporation, Norwood, MA, USA) before being pulled apart at a speed of 1 mm s^−1^ while recording the force. The maximum force *F* and the area under the curve (=work *W*) were taken as a measure for binding strength. Furthermore, photographic images were taken after the test to qualitatively assess the binding performance to the respective TVP and/or fat mimetic layer. The tests were conducted at 25 °C and at 70 °C, which is the desired core temperature of meat products [28].

### 2.5. Statistics

The assembly of the bacon-type meat analogues was performed twice with freshly prepared binder on different days. Each experiment was performed with fresh sample four times. SPSS statistics (V27, IBM Corporation, Armonk, NY, USA) was used to conduct a one-way analysis of variance (ANOVA), using the Duncan post-hoc test with a significance level of α = 5% [29].

## 3. Results & Discussion

### 3.1. Stickiness of Methylcellulose Hydrogel

First, we elucidated the stickiness of differently concentrated methylcellulose hydrogels, since adhesion and cohesion are important for binders in bacon analogues [15]. The work of adhesion as a measure of stickiness increased from 2.9 ± 0.1 mJ at a methylcellulose concentration of 2% (*w*/*w*) to a maximum of 6.0 ± 0.1 mJ at 6% (*w*/*w*) (Figure 1). At concentrations ≥ 10% (*w*/*w*), the stickiness of methylcellulose sharply decreased (Wadhesion ≤ 0.7 mJ), coinciding with a transition from cohesive to adhesive failure. At high concentrations, the sample failed to adhere to the probe because of its more pronounced solid-like character. In other words, the sample had less viscous properties that govern deformability of the adhesive and facilitate sufficient wetting of the adherend by the methylcellulose gel. Such a concentration-dependent stickiness reduction has also been reported for sugar syrups [30]. An inverted u-shaped curve for stickiness, as affected by concentration, was reported previously for pea protein–apple pectin blends [22]. The methylcellulose hydrogel with the highest stickiness, namely at 6% (*w*/*w*) (Figure 1), was chosen as binder in the tested bacon analogue system. The stickiness of the pea protein–sugar beet pectin mixture that was also used in the following experiments had been tested in another study and was in a similar range (Wadhesion = 6.0 ± 0.3 mJ) (unpublished data).

### 3.2. Binding Properties of the Binder Systems

In this part, we wanted to investigate the binding performance of the sugar beet pectin–pea protein based clean-label binder and evaluate the influence of laccase-induced solidification. Methylcellulose served as a benchmark, as it is ubiquitously used as binder in meat analogues [5]. Although the binding between the textured vegetable protein (TVP) and the fat mimetic layers is most important in bacon analogues [15], the binding strength between the same structural elements (TVP/TVP and fat mimetic /fat mimetic) is also of interest, as it allows estimation of the binding performance in other application scenarios. Furthermore, the binding strength was tested at 25 °C and 70 °C, as the product should resist collapse during consumer preparation.

The pea protein–sugar beet pectin mixture has been shown to be viscoelastic, with a certain balance between adhesion and cohesion, and was, therefore, promising for use as sticky food glue (unpublished data). It had a medium binding force between two TVP layers at 25 °C (F = 3.5 ± 1.3 N). This was significantly higher (*p* < 0.05) than between a TVP and a fat mimetic layer (F = 1.8 ± 0.8 N); however, there was no significant difference in the binding work (*p* > 0.05) (Figure 2A). The lower binding strength between the TVP and the fat mimetic may arise from the lack of adhesion of the binder to the fat mimetic, which was noticeable after the tensile strength test (red circle in Figure 3A). Both binding force and work at 70 °C were significantly lower (*p* < 0.05) than at 25 °C; however, there was no difference in binding strength for the two different structures (TVP/TVP and TVP/fat mimetic) (Figure 2D). Still, the lower adhesion of the binder to the fat mimetic layer was visible (red circle in Figure 3D). We were not able to apply the pea protein–sugar beet pectin mixture directly on the fat mimetic without causing rupture, as it was too sticky. From a practical point of view, a lower concentration of solids would lead to a lower viscosity of the mixture, allowing for direct application of the binder system on the fat mimetic. However, this would also result in lower cohesive strength and thus, most likely, binding strength [22].

The pea protein–sugar beet pectin mixture solidified upon laccase addition (unpublished data), due to crosslinking of ferulic acid present in the sugar beet pectin and tyrosine present in the used pea protein [8,25]. The solidification of the pea protein–sugar beet pectin binder upon addition of laccase thus led to a significantly (*p* < 0.05) higher binding force and work for both TVP-TVP (F = 6.3 ± 2.6 N; W = 4.3 ± 2.1 mJ) and TVP-fat mimetic layers (F = 4.0 ± 1.1 N; W = 2.0 ± 0.8 mJ) (Figure 2B) than for the ones without laccase (Figure 2A). For all pea protein–sugar beet pectin mixtures treated with laccase, clusters of solidified binder were visible after the tensile strength test (red arrows in Figure 3B), which may have contributed to increased binding strength. In addition, an improved adhesion of the solidified binder to the fat mimetic was visible, with less spots showing adhesive failure of the binder to the fat mimetic (red circle in Figure 3B), which also strengthened binding between the two structural elements. Again, the binding strength of laccase-treated pea protein–sugar beet pectin mixture decreased (*p* < 0.05) at 70 °C (Figure 2E); however, it was still higher (*p* < 0.05) compared to the pea protein–sugar beet pectin mixture without laccase addition (Figure 2D).

The methylcellulose hydrogel showed the lowest (*p* < 0.05) binding force at 25 °C among the tested binders (F = 0.5 ± 0.4 N) when it was applied between two TVP layers (Figure 2C). The binding force was higher (F = 2.0 ± 1.1 N) when a TVP layer was glued to a fat mimetic, although those differences were not significant (*p* > 0.05). This is due to the generally high standard deviations of the tensile strength test when methylcellulose hydrogels were used as binder. The methylcellulose hydrogel adhered preferably to the fat mimetic, unlike the other binders; however, it also acted more as a cohesive binder layer on its own that partly failed to adhere even to the fat mimetic after the tensile strength test (red oval in Figure 3C). The binding strength of methylcellulose between the two fat mimetic layers was equal to that of only one fat mimetic layer (Figure 2C). Interestingly, the binding strength of methylcellulose decreased (*p* < 0.05) for TVP/fat mimetic at 70 °C, while it remained high when two fat mimetic layers were glued together (Figure 2F). Furthermore, some fat mimetic from the upper layer was visible on the binder after the test, which indicated that the binding strength of methylcellulose at 70 °C was higher than the network strength of the fat mimetic in some parts (red square in Figure 3F).

### 3.3. Proposed Mechanisms

From the results of the tensile strength test (Figure 2) and the photographic images taken afterwards (Figure 3), we propose the following mechanisms underlying the binding performance:**Pea protein–sugar beet pectin mixtures had higher binding strength between TVP layers.** TVPs that are produced by high-moisture extrusion, as was the case for the material used in this study, consist mainly of proteins with hydrophilic and hydrophobic binding sites [31]. The biopolymers pea protein and sugar beet pectin used in the binder mixture are amphiphilic [25,32], facilitating a thermodynamically-driven adhesion between the binder and the TVP layer [18]. On the other hand, the used fat mimetic is a gelled emulsion, with crosslinked soy protein acting as the continuous phase [24]. The soy protein is also amphiphilic; however, the heating step in the preparation of the bacon-type analogue (see Section 2.4) led to fat melting and some fat leaking out of the dispersed phase [24,33]. Consequently, the fat mimetic became more hydrophobic, thus restricting adhesion to the binder (Figure 3A,D), in turn decreasing binding strength. In addition, high-moisture extrusion of plant protein leads to a layered and fibrous structure with irregularities and cavities [34]. The viscoelastic binder can flow into those cavities, which may result in mechanical interlocking and an increasing binding strength [17]. The used fat mimetic had a rather smooth texture. However, surface roughness studies would be necessary to assess the contribution of mechanical interlocking [19].**Laccase addition increased binding strength of pea protein–sugar beet pectin mixtures.** As mentioned above, the two structural elements, namely TVP and fat mimetic, consist of proteins that can potentially be crosslinked via tyrosine residues to the pea protein and the sugar beet pectin in the binder through laccase action. The higher binding strength (Figure 2B) and the better adhesion of the binder to the fat mimetic upon addition of laccase (Figure 3B), indicates that covalent crosslinks were built that contributed to stronger interactions between the pea protein–sugar beet pectin mixture and textured protein/fat mimetic. In general, covalent bonds are stronger than non-covalent bonds [35], which occurred between the structural elements and pea protein–sugar beet pectin mixture without laccase addition. Herz, Herz, Dreher, Gibis, Ray, Pibarot, Schmitt and Weiss [15] reported that binding between a protein extrudate and the same fat mimetic was improved when the soy-based binder was treated with transglutaminase for covalent bonds, instead of weak physical bonds only, which is in line with our results. Furthermore, an increased hardness of soy-based burger patties was achieved when the textured vegetable proteins were bound together with laccase-treated sugar beet pectin [8]. When laccase was not used, the burger patty was crumbly, and no coherent product was obtained [8]. It should be noted that the pea protein–sugar beet pectin binder should be applied in the unsolidified state, i.e., before the action of laccase action. Otherwise, the binder would be solid already and not be able to deform and penetrate the adherend. Furthermore, possible binding sites between the binder and the adherend would already be used up.**Binding strength at 25 °C was higher compared to 70 °C.** It is known that the share of viscous properties in biopolymer mixtures increases with temperature [36]. Although this can lead to an increase in adhesive strength, due to higher deformability, it is also unfavorable for cohesion. Cohesion is the strength of the binder opposing stresses that would lead to disruption [18], such as the one that is applied during the tensile strength test. In bacon-type analogues, an appropriate balance between adhesion and cohesion of the binder is critical [15], and we suggest that the increased temperature led to a lower cohesive strength of the binder.**Methylcellulose showed higher binding strength to the fat** mimetic. Methylcellulose as a hydrophobic biopolymer [13] can interact hydrophobically with the fat mimetic layer and can thus exhibit stronger interactions than with TVP. This is supported by the fact that the binding strength at 70 °C was equal to that at 25 °C when two fat mimetic layers were bound together, which was unlike the other tested binder system (Figure 2). Hydrophobic forces are known to increase with temperature [37].

## 4. Conclusions

A clean-label mixture based on pea protein and sugar beet pectin was shown to be a suitable binder for a bacon-type meat analogue, due to its ability to glue different structural elements (layers of textured vegetable protein and fat mimetic) together. The addition of laccase increased the binding strength as hypothesized, due to strong covalent interactions between the binder and the different structural elements. Overall, the biopolymer blend had a higher binding strength to textured vegetable protein when compared to methylcellulose. As such, the pea protein–sugar beet pectin system may also serve as a binder in other meat analogue products, and future studies may want to test its performance there. Subsequently, sensory analysis studies should be carried out to gain insights into its in vivo performance.

## Figures and Tables

**Figure 1 foods-12-00085-f001:**
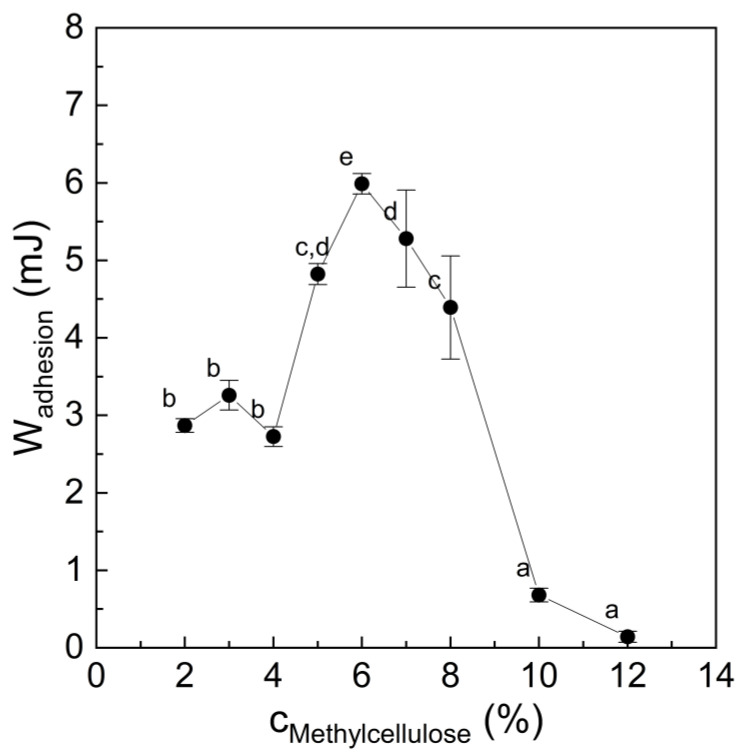
Stickiness of methylcellulose hydrogels during probe tack test, as affected by concentration. Data points with different letters denote a statistical difference (*p* < 0.05).

**Figure 2 foods-12-00085-f002:**
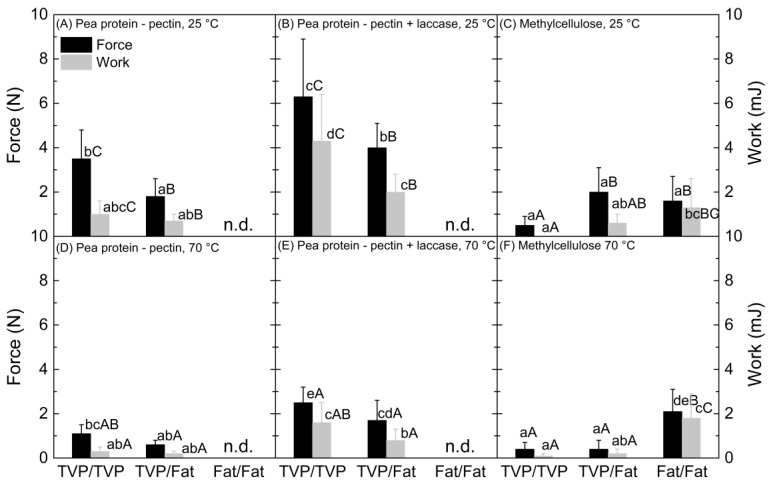
Maximum force and work during a tensile strength test of pea protein–sugar beet pectin mixtures (*r* = 2:1, 25% *w*/*w*, 100 mM NaCl) without laccase (**A**,**D**) and with laccase (**B**,**E**), and methylcellulose hydrogels (**C**,**F**) acting as binder between textured vegetable protein(s) (TVP) and/or fat mimetic (s) (Fat) at 25 °C and 70 °C. Data points with different small and capital letters denote a statistical difference (*p* < 0.05) within each temperature (impact of binder) and within each binder (impact of temperature), respectively. n.d. = not determined due to failure to apply the binder on the fat mimetic directly.

**Figure 3 foods-12-00085-f003:**
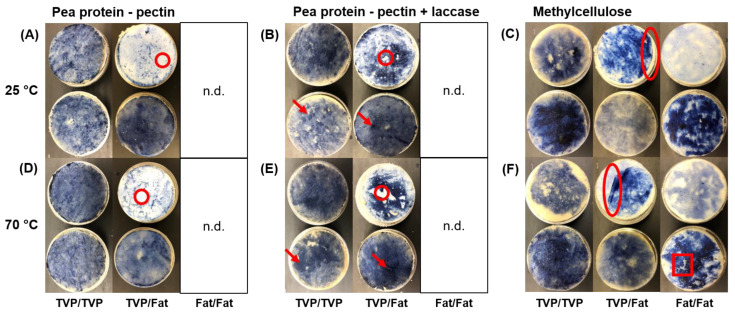
Images of pea protein–sugar beet pectin mixtures (r = 2:1, 25% *w*/*w*, 100 mM NaCl) without laccase (**A**,**D**) and with laccase (**B**,**E**), and methylcellulose hydrogels (**C**,**F**) acting as binder between textured vegetable protein(s) (TVP) and/or fat mimetic (s) (Fat) at 25 °C and 70 °C after tensile strength test. Red arrows indicate clusters of binder; red circles show failure of binder adhesion; red oval shapes indicate cohesive methylcellulose layer; and red squares show ruptured fat mimetic. Note that binder was dyed with brilliant black for better visualization. n.d. = not determined due to failure to apply the binder on the fat mimetic directly.

## Data Availability

The data used to support the findings of this study can be made available by the corresponding author upon request.

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
