# Peer review of "Comparison of Binding Properties of a Laccase-Treated Pea Protein–Sugar Beet Pectin Mixture with Methylcellulose in a Bacon-Type Meat Analogue"

_foods, 2022, doi:10.3390/foods12010085_

Round 1

Reviewer 1 Report

Some questions and remarks :

- Line 58 : "i.e. stickiness, as there. large pieces of textured vegetable proteins" : what do you mean by "as there. large" ?

- Line 104 : Avoid repetition of "control".

- Line 110 : Why was the native pH (7.4 ± 0.1) of the gels not adjusted ?

- Lines 158-159 : "... at high concentrations the sample failed to adhere to the probe because of its high cohesion" : this interpretation is inadequate. The adhesion of viscoelastic materials depend on the surface tensions of the adhesive (gel) and the adherent (probe) and on the viscoelastic properties of the adhesive. Cohesion alone is not enough to explain a lack of adhesion. So this interpretation should be corrected.

Author Response

We appreciate the time and effort spent by the reviewer to further enhance the quality of our manuscript.

- Line 58 : "i.e. stickiness, as there. large pieces of textured vegetable proteins" : what do you mean by "as there. large" ?

The punctuation mark was wrong and therefore the sentence is confusing. We have adjusted it.

- Line 104 : Avoid repetition of "control".

Done as suggested.

- Line 110 : Why was the native pH (7.4 ± 0.1) of the gels not adjusted ?

Because preliminary experiments showed no influence of pH. This information was added to the manuscript leading to the following addition: “…as we have seen in preliminary experiments that the pH had no influence on the stickiness of a methylcellulose hydrogel (data not shown).”

- Lines 158-159 : "... at high concentrations the sample failed to adhere to the probe because of its high cohesion" : this interpretation is inadequate. The adhesion of viscoelastic materials depend on the surface tensions of the adhesive (gel) and the adherent (probe) and on the viscoelastic properties of the adhesive. Cohesion alone is not enough to explain a lack of adhesion. So this interpretation should be corrected.

We agree that the explanation is lacking accuracy. Certainly, the surface tension of the adhesive and the adherent govern adhesion, but the surface tension is likely to be similar regarding the methylcellulose concentration. Therefore, it is rather the viscoelastic properties and the lack of deformability that is responsible for adhesive failure. We have added a more detailed explanation leading to the following change: “At high concentrations the sample failed to adhere to the probe because of its more pronounced solid-like character. In other words, the sample had less viscous properties that govern deformability of the adhesive and facilitates sufficient wetting of the adherend by the methylcellulose gel.”

Reviewer 2 Report

This manuscript is about the binding properties of two different ingredients in meat analogue development. The topic is relevant and it addresses a specific gap in the field. Compared with other published material, this manuscript studied newer types of binder. The methodology is fine, and the conclusions are consistent with the evidence and arguments presented. The references are appropriate. The tables and figures are fine. Some specific comments are as following:

·         Please describe the intention of targeting bacon analogues?

·         Need to mention the other major ingredients used for the formulation of target product.

·         Line No. 104: “A control sample without laccase addition served as control” reconstruct the sentence.

·         Line No.107: What should be the temperature of water?

·         Line No. 148: “Duncan 148 post-hoc test with a significance level of α = 5 %”, add reference.

Author Response

This manuscript is about the binding properties of two different ingredients in meat analogue development. The topic is relevant and it addresses a specific gap in the field. Compared with other published material, this manuscript studied newer types of binder. The methodology is fine, and the conclusions are consistent with the evidence and arguments presented. The references are appropriate. The tables and figures are fine.

We appreciate the general positive opinion of the reviewer and would like to express our gratitude.

Some specific comments are as following:

- Please describe the intention of targeting bacon analogues?

We have added more information to the introduction to underline the interest in bacon analogues in this study. This led to the following changes: “Taken together, the binding mechanism in the popular product category of bacon type meat analogues functions different as compared to other meat analogue products. Therefore, a bacon type meat analogue is an interesting model matrix to study the suitability of using a sticky food glue instead of methylcellulose that binds through gelling.”

- Need to mention the other major ingredients used for the formulation of target product.

Done as request in the introduction: “Furthermore, flavor and color components may be added to imitate sensorial appearance of the original.”

- Line No. 104: “A control sample without laccase addition served as control” reconstruct the sentence.

Corrected

- Line No.107: What should be the temperature of water?

The temperature was room temperature and the information was added to the manuscript

- Line No. 148: “Duncan 148 post-hoc test with a significance level of α = 5 %”, add reference.

Done as requested.

Reviewer 3 Report

Manuscript Number: foods-2070553

Manuscript Title: Comparison of binding properties of a pea protein – sugar beet pectin mixture with methylcellulose in a bacon type meat analogue

This paper addresses an important and interesting problem about the stickiness of pea protein - pectin mixtures. The authors investigated the binding properties of a pea protein – sugar beet pectin mixture with laccase and methylcellulose in a bacon type meat analogue. In general, I think this article  need minor revisions.

 Comments:

1. It is suggested that the title should be added “laccase”.

2. In the Materials and Methods section, it is suggested that the determination method of the images shown in Figure 3 should be supplemented.

Author Response

This paper addresses an important and interesting problem about the stickiness of pea protein - pectin mixtures. The authors investigated the binding properties of a pea protein – sugar beet pectin mixture with laccase and methylcellulose in a bacon type meat analogue. In general, I think this article  need minor revisions.

We thank the reviewer for his/her positive opinion.

 Comments:

  1. It is suggested that the title should be added “laccase”.

We agree and changed the title to: “Comparison of binding properties of a laccase-treated pea protein – sugar beet pectin mixture with methylcellulose in a bacon type meat analogue”

  1. In the Materials and Methods section, it is suggested that the determination method of the images shown in Figure 3 should be supplemented.

This was changed to: “…after the test to qualitatively assess the binding performance to the respective TVP and/or fat mimic layer.”

Reviewer 4 Report

This was studied to Comparison of binding properties of a pea protein-sugar beet pectin mixture with methylcellulose in a bacon type meat analogue, there are some studies have reported the similar papers. However, The paper is so simple and lacks enough data to support its result and “Proposed mechanisms”.

1. Title is not include the content of this paper, need improve.

2. Abstract The content is disordered, need to rewritten. 

3. Introduction is so simple, and added some information about the application in a bacon type meat products.

4. L 49-56 The sentences need improve, the content in the study of Kim, et al. [14] is not relate with the aim of paper.

5. The aim of this paper is not clear.

6. More information of Raw materials are provided, the information of Raw materials is not clear.

7. 2.2 Preparation of binders is not clear, it is important to readers, please rewrite.

8. 4. 2.5 Statistics please rewrite.

9. 3.1 Stickiness of methylcellulose hydrogel   The structure of the section is irregular, lacked of hierarchy, please rewrite.

11. 6. 3.3 Proposed mechanisms  The section lacks enough data to support.

12. The figures need improve.

Author Response

This was studied to Comparison of binding properties of a pea protein-sugar beet pectin mixture with methylcellulose in a bacon type meat analogue, there are some studies have reported the similar papers. However, The paper is so simple and lacks enough data to support its result and “Proposed mechanisms”.

We appreciate the time and effort spent by the reviewer to further enhance the quality of our manuscript.

  1. Title is not include the content of this paper, need improve.

We have changed the title so that it contains the term laccase. We think that this title now describes the content of the manuscript properly: “Comparison of binding properties of a laccase-treated pea protein – sugar beet pectin mixture with methylcellulose in a bacon type meat analogue”

  1. Abstract The content is disordered, need to rewritten. 

We respectfully disagree. The abstract follows the generally recommended and accepted form for an abstract, namely introduction, methodology, results and discussion, and has been positively commented on by other reviewers. It provides the reader with a short version of the content of the article. It would be useful if the reviewer could specify which parts seem to be disordered.

  1. Introduction is so simple, and added some information about the application in a bacon type meat products.

We have specified why bacon was chosen as model matrix and what the intention of the used binding system was. This led to the following change to the manuscript: “Taken together, the binding mechanism in the popular product category of bacon type meat analogues functions different as compared to other meat analogue products. Therefore, a bacon type meat analogue is an interesting model matrix to study the suitability of using a sticky food glue instead of methylcellulose that binds through gelling.”

  1. L 49-56 The sentences need improve, the content in the study of Kim, et al. [14] is not relate with the aim of paper.

We have rephrased the sentences to make them clearer for the reader. Furthermore, the reference by Kim, et al. was removed.

  1. The aim of this paper is not clear.

An aim was added to the end of the introduction part: “we aimed to find a novel, clean-label binding agent that can replace methylcellulose in commercial applications.”

  1. More information of Raw materials are provided, the information of Raw materials is not clear.

The information currently given allows other researchers to replicate the experiments, and provides general information on the materials used. Again it would be helpful to know what specifically is not clear there to the reviewer.

  1. 2.2 Preparation of binders is not clear, it is important to readers, please rewrite.

More detailed information was provided in the section leading to several changes to the paragraph:

“A previously reported method [23] was used to prepare a viscoelastic food glue that was based on pea protein and sugar beet pectin. The pea protein to sugar beet pectin weight ratio was 2:1, the total biopolymer concentration (= pea protein & sugar beet pectin) 25% (w/w), and 100 mmol of NaCl was added. The pH was adjusted to 6.0 with NaOH as this was shown to promote stickiness and lies within the pH optimum for laccase [27]. A third of the required water at room temperature was used for dissolving 100 nkat laccase per gram solid substance (25% w/w) [28] that was then added to the mixture and incorporated into it by stirring vigorously by hand until a homogeneous mass was obtained. This happened right before using the binder for assembly of the bacon type meat analogue (see section 2.4). A sample without laccase addition served as control.

Methylcellulose hydrogels were prepared by weighing 2-12% (w/w) methylcellulose powder in water and stirring it in a food processor (UMC 5, Stephan, Diessen, Germany) at medium speed for 10 min. Vacuum was applied during stirring with a diaphragm vacuum pump (MZ 2C, Vacuubrand, Wertheim, Germany) to avoid the incorporation of air bubbles. The native pH (7.4 ± 0.1) of the gels was not adjusted as we have seen in preliminary experiments that the pH had no influence on the stickiness of a methylcellulose hydrogel (data not shown).

For all binders, sodium azide (0.02% w/w) was added to prevent microbial growth during storage and brilliant black (0.01 % w/w) was added to dye the binder for better visualization.”

  1. 4. 2.5 Statistics please rewrite.

Done as requested: “The assembly of the bacon type meat analogues was done twice with freshly prepared binder on different days. Each experiment was performed with fresh sample four times.”

  1. 3.1 Stickiness of methylcellulose hydrogel   The structure of the section is irregular, lacked of hierarchy, please rewrite.

The section explains the concentration-dependent stickiness of methylcellulose gels from low to high concentration thereby providing an explanation of the observed phenomena. We would rewrite the section if the reviewer proposed a structure that suits him/her and the reader better.

  1. 6. 3.3 Proposed mechanisms  The section lacks enough data to support.

Each explanation is backed with data and qualitative insights from the experiments. Furthermore, state of the art knowledge regarding those matters from literature is included to further support the proposed mechanisms. A statistical analysis has been carried out. As such, the proposed mechanism provides a valid explanation of what the underlying reason for the observed results are, and provides a good base for further discussions within the scientific community. In our opinion, that is what the discussion section of a research paper should do.

  1. The figures need improve.

Again, we would highly appreciate more concrete suggestions as to where and what kind of improvements are necessary. Is it the quality of images? The visualization of data? The legends?

Round 2

Reviewer 4 Report

Although the article has been revised, it is still difficult to meet the requirements.

The other, please the authors keep an optimistic attitude to reviewer, our work is to improve your paper.

1. The abstract part is just a simple description of the results of the paper, without using data to explain. In addition, the author's introduction to the experimental method is not perfect.

2. More information of Raw materials are provided, such as protein content about High-moisture textured vegetable protein; Carbohydrate content in Pea protein, the factors influence the water holding capacity and other properties.

3. Brief description of methods about Dreher, et al. [25], [26] and [27].

4. The formation of figures need improve, such as Figure 1 and L 184 and others.

Author Response

Although the article has been revised, it is still difficult to meet the requirements.

The other, please the authors keep an optimistic attitude to reviewer, our work is to improve your paper.

We thank the reviewer for her/his time and contributions to further improve our manuscript. We were just lacking some further information as to where exactly corrections are necessary to address the reviewers concerns better, which is why we have asked for that. Apologies if this came off the wrong way.

  1. The abstract part is just a simple description of the results of the paper, without using data to explain. In addition, the author's introduction to the experimental method is not perfect.

We agree that concrete results would help to explain the results of the manuscript better. Furthermore, we have given more information on the experimental methods. This led to a largely revised abstract:

“A bacon type meat analogue consists of different structural layers such as textured protein and fat mimic. To obtain a coherent and appealing product, a suitable binder must glue those elements together. A mixture based on pea protein and sugar beet pectin (r = 2:1, 25% w/w solids, pH 6) with and without laccase addition and a methylcellulose hydrogel (6% w/w) serving as benchmark were applied as binder between textured protein and fat mimic. A tensile strength test during which the layers were torn apart was performed to measure the binding ability. The pea protein-sugar beet pectin mixture without laccase was viscoelastic and had medium and low binding strength at 25 (F £ 3.5 N) and 70 °C (F » 1.0 N), respectively. The addition of laccase solidified the mixture and increased binding strength at 25 (F ³ 4.0 N) and 70 °C (F » 2.0 N) due to covalent bonds within the binder and between the binder and the textured protein or the fat mimic layers. Generally, the binding strength was higher when two textured protein layers were glued together. The binding properties of methylcellulose hydrogel was low (F £ 2.0N) except when two fat mimic layers were bound due to hydrophobic interactions becoming dominant. The investigated mixed pectin – pea protein system is able serve as a clean-label binder in bacon type meat analogues and the application in other products seems promising.“

  1. More information of Raw materials are provided, such as protein content about High-moisture textured vegetable protein; Carbohydrate content in Pea protein, the factors influence the water holding capacity and other properties.

We are very sorry to say that the protein content of the textured vegetable protein was not shared with us due to confidentiality. We have, however, provided more information for the pea protein.

  1. Brief description of methods about Dreher, et al. [25], [26] and [27].

Done as suggested leading to the following addition: “In brief, the soy protein isolate was thoroughly mixed with a lab rotor-stator homogenizer, adjusted to pH 7 and rapeseed oil was slowly added while homogenization continued. Then, transglutaminase was added, the mass was filled in casings, and incubated at 40 °C for 60 min to enable protein crosslinking. Afterwards, the mass was heated to a core temperature of 85 °C for microbial safety and stored at 2 °C until further usage.“

  1. The formation of figures need improve, such as Figure 1 and L 184 and others.

We have improved the arrangement of figures.
